# Global inventory of doubly substituted isotopologues of methane ($\Delta^{13}CH_3D$ and $\Delta^{12}CH_2D_2$)

**Authors:** Sara M. Defratyka[1,2], Julianne M. Fernandez[3, *], Getachew A Adnew[4], Guannan Dong[5], Peter M.J. Douglas[6], Daniel L. Eldridge[7], Giuseppe Etiope[8,9], Thomas Giunta[10], Mojhgan A. Haghnegahdar[3], Alexander N. Hristov[11], Nicole Hultquist[3], Iñaki Vadillo[12], Josue Jautzy[13], Ji-Hyun Kim[14], Jabrane Labidi[15], Ellen Lalk[16,**], Wil Leavitt[17,18,19], Jiawen Li[18], Li-Hung Lin[20], Jiarui Liu[21,22], Lucía Ojeda[12], Shuhei Ono[16], Jeemin H. Rhim[23], Thomas Röckmann[24], Barbara Sherwood Lollar[25], Malavika Sivan[24], Jiayang Sun[3,26], Gregory T. Ventura[27], David T. Wang[16,***] , Edward D. Young[22], Naizhong Zhang[28] and Tim Arnold[29,1]

[1] School of GeoSciences, University of Edinburgh, Edinburgh, UK

[2] National Physical Laboratory, Hampton Road, London, UK

[3] Department of Geology, University of Maryland, College Park; College Park, Maryland, USA

[4] Terrestrial Ecosystem Analysis Group, Institute of Geography, People and Processes, Department of Geosciences and Resource management, Science Faculty, University of Copenhagen, Øster Voldgade 10 1350 Copenhagen K, Copenhagen, Denmark

[5] Division of Geological and Planetary Sciences, California Institute of Technology, Pasadena, CA 91125, USA

[6] Earth and Planetary Sciences and Geotop Research Centre, McGill University, Montreal, QC, Canada

[7] Earth and Environmental Sciences Division, Los Alamos National Laboratory, Los Alamos, New Mexico 87545, USA

[8] Istituto Nazionale di Geofisica e Vulcanologia, Sezione Roma 2, Rome, Italy

[9] Faculty of Environmental Science and Engineering, Babes-Bolyai University, Cluj-Napoca, Romania

[10] Geo-Ocean (UMR6538), University Brest, CNRS, Ifremer, Plouzané, France

[11] Department of Animal Science, The Pennsylvania State University

[12] Department of Geology, Faculty of Science, University of Malaga, Malaga, Spain

[13] Geological Survey of Canada - Natural Resources Canada

[14] Marine Geology & Energy Division, Korea Institute of Geoscience and Mineral Resources, Daejeon, South Korea

[15] Université de Paris, Institut de Physique du Globe de Paris, CNRS, F-75005 Paris, France

[16] Department of Earth, Atmospheric, and Planetary Sciences, Massachusetts Institute of Technology, 77 Massachusetts Ave, Cambridge, MA 02139 USA

[17] Department of Geology & Geophysics, University of Utah, Salt Lake City, UT, USA

[18] Department of Earth Sciences, Dartmouth College, Hanover, NH, USA

[19] Department of Chemistry, Dartmouth College, Hanover, NH, USA

[20] Department of Geosciences, National Taiwan University, Taipei 106, Taiwan

[21] Marine Science Institute, University of California, Santa Barbara, California 93106, USA

[22] Department of Earth, Planetary, and Space Sciences, University of California, Los Angeles, California 90095, USA

[23] Department of Ecology, Evolution, and Marine Biology University of California, Santa Barbara

[24] Institute for Marine and Atmospheric research Utrecht (IMAU), Utrecht University, Utrecht, the Netherlands

[25] Department of Earth Sciences, 22 Ursula Franklin St. University of Toronto, Toronto, ON Canada M5S 3B1

[26] Air Resources Laboratory, National Oceanic and Atmospheric Administration, College Park, Maryland 20740, USA

[27] Department of Geology, Saint Mary's University, Halifax, NS B3H 3C3, Canada

[28] Laboratory for Air Pollution / Environmental Technology, Empa, 8600 Dübendorf, Switzerland

[29] Department of Physical Geography and Ecosystem Science, Lund University, Sweden

[*] Now at: Global Monitoring Laboratory, National Oceanic and Atmospheric Administration, Boulder, CO, USA,

[**] Now at: U.S. Geological Survey, Woods Hole Coastal and Marine Science Center, Woods Hole, MA 02543, USA

[***] Now at: U.S. Department of Energy, 1000 Independence Ave SW, Washington, DC 20585 USA

*Correspondence to:* Sara M. Defratyka (sara.defratyka@ed.ac.uk)

**Short summary (up to 500 characters).** Measurement of methane's doubly substituted isotopologues at natural abundances holds promise for better constraining the Earth's atmospheric $CH_4$ budget. We compiled 1475 measurements from field samples and laboratory experiments, conducted since 2014, to facilitate the differentiation of $CH_4$ formation pathways and processes, to identify existing gaps limiting application of $\Delta^{13}CH_3D$ and $\Delta^{12}CH_2D_2$, and to develop isotope ratio source signature inputs for global $CH_4$ flux modelling.

**Abstract:**

Measurements of methane ($CH_4$) molecules containing two rare isotopes ($^{13}CH_3D$ and $^{12}CH_2D_2$), also termed doubly substituted or 'clumped' isotopologues, have the potential to provide two additional isotopic dimensions to help investigate the mechanisms underlying global atmospheric trends in $CH_4$. In this work, we summarise the current state of research on doubly substituted $CH_4$ isotopologues, with an emphasis on compiling results of all relevant work. The database comprises 1475 records compiled from the literature published until April 2025 (https://dx.doi.org/10.5285/51ae627da5fb41b8a767ee6c653f83e6). For field samples, 40% of records were sourced from natural gas reservoirs, while microbial terrestrial (e.g., agriculture, lake, wetland) samples account only for 12.5%. Lakes samples contribute 75% to collected microbial terrestrial samples. There is limited or no representation of samples coming from significant microbial $CH_4$ sources to the atmosphere, like wetlands, agricultural practices and landfills. To date, laboratory experiments were mostly focused on microbial (28% of samples from laboratory experiments) and pyrogenic (15%) methanogenesis or anaerobic (16%), and aerobic (8%) $CH_4$ oxidation, with only single study of photochemical oxidation via OH and Cl, which constitutes 5% of the laboratory experiments entries. The distinct ranges of $\Delta^{13}CH_3D$ and $\Delta^{12}CH_2D_2$ values measured in these studies suggests their potential to improve our understanding of atmospheric $CH_4$. This work provides an overview of the major gaps in measurements and identifies where further studies should be focussed to enable the highest impact on understanding global $CH_4$.

## 1. Introduction

Methane's bulk isotopic signatures (in particular $\delta^{13}C\text{-}CH_4$), have been commonly used to constrain $CH_4$ emissions sources and budget changes (Basu et al., 2022; Lan et al., 2021; Menoud et al., 2022a; Sherwood et al., 2017; Turner et al., 2019). While the observed recent negative trend in $\delta^{13}C\text{-}CH_4$ with increasing $CH_4$ mole fraction in the atmosphere implies a shift towards increasing microbial sources, the magnitude of this shift is difficult to quantify owing to the uncertainty in the isotopic source terms (Nisbet et al., 2019). Thus, additional independent tracers of $CH_4$ fluxes to the atmosphere would be useful to improve the understanding of global $CH_4$ changes.

The isotopologues $^{13}CH_3D$ and $^{12}CH_2D_2$, referred to as doubly-substituted or "clumped" isotopologues, are thermodynamically more stable than the more abundant singly substituted $CH_4$ (i.e., $^{13}CH_4$ and $^{12}CH_3D$). High precision measurements of the ratios of these rarer isotopologues present new tracer capabilities to quantify $CH_4$ sources and sinks (e.g., Douglas et al., 2017; Eiler, 2007; Haghnegahdar et al., 2017; Sivan et al., 2024; Stolper et al., 2014b; Young et al., 2017). The reported values, $\Delta^{13}CH_3D$ and $\Delta^{12}CH_2D_2$, represent the measured isotopologue ratios ($^{13}CH_3D/^{12}CH_4$ and $^{12}CH_2D_2/^{12}CH_4$, respectively) relative to their calculated values that assumes a random distribution of isotopes amongst the $CH_4$ isotopologues. This parameterization proves beneficial, as at thermodynamic isotopic equilibrium, the deviation in these isotopologue ratios from a purely random distribution is solely a function of temperature and it is independent from the bulk isotopic contents. Therefore, measurements of $\Delta^{13}CH_3D$ and $\Delta^{12}CH_2D_2$ can constrain $CH_4$ formation temperatures, if the $CH_4$ has formed in thermodynamic isotopic equilibrium. An important aspect of this parameterization is that at

sufficiently high temperatures under thermodynamic isotopic equilibrium (where exchange of isotopes between isotopologues is fully reversible) the doubly substituted isotopic signature tends towards zero. At low temperatures, however, the abundance of clumped isotopes is much higher than expected from random distribution (e.g., Eldridge et al., 2019; Stolper et al., 2014a; Young et al., 2016).

When $CH_4$ is not in thermodynamic isotopic equilibrium, values of $\Delta^{13}CH_3D$ and $\Delta^{12}CH_2D_2$ can reflect other physicochemical processes, such as their formation and consumption reactions (kinetic isotope effects, combinatorial effects, etc.), mixing of different sources, and physical transport processes such as molecular diffusion (e.g., Douglas et al., 2017; Gonzalez et al., 2019; Ono et al., 2014; Röckmann et al., 2016; Stolper et al., 2014b; Wang et al., 2024; Yeung, 2016; Young, 2019; Young et al., 2017). Therefore, measurements of doubly substituted isotopologues can provide additional analytical dimensions to distinguish between atmospheric sources (e.g., microbial, thermogenic, and abiotic $CH_4$) and sinks (Chung and Arnold, 2021; Douglas et al., 2017; Haghnegahdar et al., 2017; Stolper et al., 2014a; Whitehill et al., 2017; Young, 2019). For example, it is currently understood that the $\Delta^{13}CH_3D$ of atmospheric $CH_4$ is more sensitive to sources than sinks because it does not appear to be strongly affected by currently known sink reactions, while $\Delta^{12}CH_2D_2$ is currently understood to be sensitive to both atmospheric $CH_4$ sources and sinks (Chung and Arnold, 2021; Haghnegahdar et al., 2017, 2023, 2024; Sivan et al., 2024; Whitehill et al., 2017). Thus, the atmospheric monitoring of $\Delta^{13}CH_3D$ and $\Delta^{12}CH_2D_2$ has the potential to yield novel and unique insights into the temporal and spatial variations in atmospheric $CH_4$ source and sink reactions.

The first attempt to measure the rare $CH_4$ isotopologues from the ambient air was presented by Mroz et al. (1989), with further methods development refined by Ma et al. (2008) and Tsuji et al. (2012). The first precise measurements of doubly substituted $CH_4$, specifically $\Delta_{18}$ (combined $\Delta^{13}CH_3D$ and $\Delta^{12}CH_2D_2$) or $\Delta^{13}CH_3D$ were published in 2014 (Ono et al., 2014; Stolper et al., 2014a, b). Young et al., (2017) reported on the first $^{12}CH_2D_2$ data from laboratory and natural $CH_4$ sources. Since then, these measurements have become more relevant, particularly within the isotope geochemistry community. Measuring $\Delta^{13}CH_3D$ and $\Delta^{12}CH_2D_2$ from ambient air samples, however, is more challenging as it requires the collection and quantitative extraction of $CH_4$ from about 1000 L of air (1 $m^3$). The first $\Delta^{13}CH_3D$ and $\Delta^{12}CH_2D_2$ measurements from the atmosphere, based on ambient air collections in Maryland (USA) and Utrecht (Netherlands), differed from model predictions of the atmosphere based on certain assumptions of source and sink reaction signatures (Chung and Arnold, 2021; Haghnegahdar et al., 2017, 2023; Sivan et al., 2024). The discrepancy could therefore come from either incorrectly assigned kinetic isotope effects associated with sink reactions or the assumptions regarding source signatures, or both (Haghnegahdar et al., 2023; Sivan et al., 2024; Wang et al., 2023b). This underscores the importance of obtaining improved constraints on source signatures and the isotope effects associated with sink reactions for improving the utility of $\Delta^{13}CH_3D$ and $\Delta^{12}CH_2D_2$ in the study of atmospheric $CH_4$.

For this study, we have compiled an open-source database (Defratyka et al., 2025) (https://dx.doi.org/10.5285/51ae627da5fb41b8a767ee6c653f83e6) of existing measurements of $\Delta^{13}CH_3D$ and $\Delta^{12}CH_2D_2$, including studies where only $\Delta^{13}CH_3D$ was measured, from peer-reviewed scientific journal publications. The database contains almost 1500 values of doubly substituted isotope ratio measurements, from about 75 peer-reviewed scientific publications. The database is designed for utilization by the geochemistry and atmospheric science communities. This paper describes the collected $\Delta^{13}CH_3D$ and $\Delta^{12}CH_2D_2$ values that are included in the database. Our purpose is to present the current knowledge of doubly substituted isotopologues of $CH_4$ and identify existing gaps that presently limit our ability to apply $\Delta^{13}CH_3D$ and $\Delta^{12}CH_2D_2$ to understanding of atmospheric $CH_4$.

## 2. $CH_4$ doubly substituted isotopologue ratios

### 2.1. $\Delta^{13}CH_3D$ and $\Delta^{12}CH_2D_2$ notations and calibration

A comprehensive review of the theory and nomenclature of doubly substituted isotopologue geochemistry is detailed in Eiler (2007, 2013), Wang et al., (2004) and Young et al. (2016, 2017). Briefly, doubly substituted isotopologue ratios of $CH_4$ are reported and parameterized as $\Delta^{13}CH_3D$ and $\Delta^{12}CH_2D_2$ values, defined to quantify a measured difference in the isotopologue ratios relative to a random distribution:

$$145 \qquad \Delta^{13}CH_3D = \frac{R_{sample}^{^{13}CH_3D}}{R_{stochastic}^{^{13}CH_3D}} - 1 \qquad\qquad (1),$$

$$\Delta^{12}CH_2D_2 = \frac{R_{sample}^{^{12}CH_2D_2}}{R_{stochastic}^{^{12}CH_2D_2}} - 1 \qquad\qquad (2).$$

Where:

$R_{sample}^{^{13}CH_3D}$ and $R_{sample}^{^{12}CH_2D_2}$ are the measured isotopologue ratios of $^{13}CH_3D / ^{12}CH_4$ and $^{12}CH_2D_2 / ^{12}CH_4$, respectively,

and

$R_{stochastic}^{^{13}CH_3D}$ and $R_{stochastic}^{^{12}CH_2D_2}$ are the calculated isotopologue ratios of $^{13}CH_3D / ^{12}CH_4$ and $^{12}CH_2D_2 / ^{12}CH_4$, respectively, based on the assumption of a random distribution of isotopes amongst all stable isotopologues.

As an effect, the isotopologue ratio approaches that based on a random distribution under high-
temperature equilibrium conditions, which by definition results in $\Delta^{13}CH_3D$ or $\Delta^{12}CH_2D_2$ values of zero (e.g., Douglas et al., 2016; Eiler, 2007, 2013; Stolper et al., 2014a; Young, 2019). It should be noted that non-zero values of $\Delta^{13}CH_3D$ or $\Delta^{12}CH_2D_2$ can result from the simple mixing of two separate $CH_4$ pools with distinct bulk isotopic compositions, without any chemical or physical processes inducing isotopic fractionation (e.g., Young et al., 2016).

In this paper, the terms 'enriched' and 'depleted' refer to comparative values of $\Delta^{13}CH_3D$ or $\Delta^{12}CH_2D_2$ – higher numbers as enriched and lower numbers as depleted – for example when comparing samples of $CH_4$, products and reactants of a chemical reaction, or the evolution of $CH_4$ in a physical process.

### 2.2. Existing instrumentation

The measurement of $\Delta^{13}CH_3D$ and $\Delta^{12}CH_2D_2$ is resource intensive, requiring specialised facilities that
are currently not widely available (e.g., Eiler, 2007; Liu et al., 2024b; Ono et al., 2014a; Sivan et al., 2024; Stolper et al., 2014a; Young et al., 2017). Magnetic sector High Resolution Isotope Ratio Mass Spectrometry (HR-IRMS) is the most common method to measure $\Delta^{13}CH_3D$ and $\Delta^{12}CH_2D_2$ (Dong et al., 2020; Eldridge et al., 2019; Haghnegahdar et al., 2023; Liu et al., 2024b; Sivan et al., 2024; Stolper et al., 2014a; Sun et al., 2023; Thiagarajan et al., 2020; Wang et al., 2023a; Young et al., 2016, 2025; Zhang
et al., 2021). The first magnetic sector HR-IRMS instrument developed for this purpose was the non-commercial prototype model of the Thermo Scientific 253 Ultra HR-IRMS (developed and installed solely at the California Institute of Technology) that was able to measure a value of the combined $^{13}CH_3D$ and $^{12}CH_2D_2$ abundances via a parameter defined as $\Delta_{18}$ (Eiler et al., 2013; Stolper et al., 2014 a,b; Stolper et al., 2015). A large-radius gas-source multiple-collector isotope ratio mass spectrometer
capable of operating up to a mass resolving power (MRP) of 80,000 (Panorama, Nu Instrument) was the first developed HR-IRMS to measure separately $\Delta^{13}CH_3D$ and $\Delta^{12}CH_2D_2$ (Young et al., 2016, 2017). This was followed by the commercially-available production model of the Thermo Scientific Ultra HR-

IRMS that can also measure $\Delta^{13}CH_3D$ and $\Delta^{12}CH_2D_2$ and routinely achieves a MRP of 30-35,000 (e.g., Eldridge et al., 2019; Thiagarajan et al., 2020; Zhang et al., 2021; Wang et al. 2023a; Sivan et al., 2024). The obtained MRP allows to achieve precise measurements for sample of >2 mL STP (standard temperature and pressure) of $CH_4$ (~ 80 µmol) for Panorama (e.g., Labidi et al., 2020) and 3 ± 1 mL STP for Ultra (Sivan et al., 2024). Measurements of smaller volume of $CH_4$ sample result in larger uncertainties caused by degraded counting statistic. The detailed description of the performance of these instruments and measurement protocols for different laboratories can be found in the cited references above.

Distinct from mass spectrometry, measurements of $\Delta^{13}CH_3D$ and $\Delta^{12}CH_2D_2$ are also possible owing to developments in infrared absorption spectroscopy using quantum cascade lasers (TILDAS, Aerodyne Research) operated in near room temperature with narrow line widths and high power (Chen et al., 2022; Gonzalez et al., 2019; Ono et al., 2014; Prokhorov and Mohn, 2022; Zhang et al., 2025). The first TILDAS instrument to achieve high precision $\Delta^{13}CH_3D$ measurements was demonstrated at the Massachusetts Institute of Technology in 2014 (Ono et al., 2014). $\Delta^{13}CH_3D$ measurement by the TILDAS instrument are achieved using the absorption line in a spectral region around 8.6 µm, as there are fewer interferences from hot bands (Ono et al., 2014). Gonzalez et al. (2019) presented a possibility to implement TILDAS to measure $\Delta^{12}CH_2D_2$ with precision of 0.5 ‰. Routinely, TILDAS measurements requires 10 mL of $CH_4$ for $\Delta^{13}CH_3D$ measurements and 20 mL for $\Delta^{12}CH_2D_2$ (e.g., Gonzalez et al., 2019; Ono et al., 2014). Recently, Zhang et al. (2025) were able to reduce the required volume of $CH_4$ to 3-7 mL STP for $\Delta^{13}CH_3D$ and to 10 mL STP for $\Delta^{12}CH_2D_2$, via further instrument optimization.

HR-IRMS signal stability of the detected ions at very low ion currents is key to enable precise isotope ratio measurement through signal acquisition over several hours or even days (e.g., Sivan et al., 2024; Stolper et al., 2014a; Young et al., 2016). Across instrumentation, internal precision and external reproducibility are comparable between laboratories and instruments, achieving down to 0.35 ‰ for $\Delta^{13}CH_3D$ and 1.35 ‰ for $\Delta^{12}CH_2D_2$, depending on the measurement technique. The TILDAS and Panorama systems were cross-calibrated on the same set of carbon and hydrogen isotopically characterised laboratory working standards for $CH_4$ to ensure accuracy between different analytical systems (Ono et al., 2014; Young et al., 2017; Zhang et al., 2025).

At thermodynamic isotopic equilibrium, $\Delta^{13}CH_3D$ and $\Delta^{12}CH_2D_2$ values can be linked to a $CH_4$ formation temperature via monotonic functions, presented in Table S1 (Beaudry et al., 2021; Douglas et al., 2017; Eldridge et al., 2019; Gruen et al., 2018; Liu and Liu, 2016; Ono et al., 2014; Stolper et al., 2014a; Thiagarajan et al., 2020; Webb and Miller, 2014; Young et al., 2017; Zhang et al., 2021). Different theoretical calculations have been used to obtain these relationships but discrepancies among them are smaller than the current analytical uncertainties. Currently, equilibrated gas experiments along with these theoretical calculations are the basis for calibrating $\Delta^{13}CH_3D$ and $\Delta^{12}CH_2D_2$ measurements via either magnetic sector HR-IRMS or laser spectroscopy (Eldridge et al., 2019; Liu et al., 2024b; Ono et al., 2014; Sivan et al., 2024; Stolper et al., 2014a; Wang et al., 2015).

### 2.2.1. Samples extraction and purification

Quantitative extraction and complete purification of $CH_4$ from natural samples is currently necessary to attain the required precision and accuracy to detect differences in clumped isotopic composition (Eiler, 2007; Prokhorov and Mohn, 2022; Safi et al., 2024; Sivan et al., 2024; Sun et al., 2023; Young et al., 2017). Two main methods have been applied so far across laboratories. One employs cryogenic trapping at near absolute zero temperature using a Helium cryostat (Stolper et al., 2014a) and the other have used chromatographic separations techniques (Young et al., 2017).

Measuring doubly substituted isotopologues in ambient air is a major analytical challenge. Since krypton has a similar concentration in the atmosphere and boiling point as $CH_4$ (Kr: 1.14 ppm in the atmosphere, -153.4 °C boiling point; $CH_4$: 1.93 ppm, -161.5 °C), it makes separation by fractional distillation alone impossible. Recently, combined gas chromatography and cryogenic methods were successfully implemented to purify $CH_4$ from $10^2$-$10^3$ litres of ambient air to measure both $\Delta^{13}CH_3D$ and $\Delta^{12}CH_2D_2$. These approaches generally involve the pumping of large volumes of air through sequential cryogenic traps that selectively isolate $CH_4$ from other contaminants using established absorbents (Haghnegahdar et al., 2023; Sivan et al., 2024).

## 3. Database methods and description

### 3.1. Data gathering

The compilation of this doubly substituted $CH_4$ isotopologues database is inspired by similar efforts of existing databases for bulk isotopes of $CH_4$ (Lan et al., 2021; Menoud et al., 2022a, b; Sherwood et al., 2017, 2021). To verify if the compiled data compares well with previous studies, figure 1 and table 1 present bulk isotopes from this database in the reference to previously reported $\delta^{13}C$-$CH_4$ and $\delta D$-$CH_4$ (Menoud et al., 2022a; Sherwood et al., 2021). Across compared group types, our additional bulk isotope ratio data fall within the established ranges. Fossil fuel and thermogenic source signatures overlap, however, they are not strictly equivalent. Thermogenic $CH_4$ in our dataset is slightly enriched ($\delta^{13}C$-$CH_4$: −39.0 ± 9.6‰; $\delta D$-$CH_4$: −169.2 ± 41.9‰), compared to fossil fuel. For the comparison, only terrestrial microbial (e.g., agriculture, lakes, wetlands) from this database is compared with previously compiled data and shows strong agreement with the range of previous microbial samples, with depleted $\delta^{13}C$-$CH_4$ and $\delta D$-$CH_4$ values ($\delta^{13}C$-$CH_4$: −62.9 ± 13.2‰; $\delta D$-$CH_4$: −298.1 ± 47.7‰). Pyrogenic methane, though represented by only two samples in the new database, shows $\delta^{13}C$-$CH_4$ and $\delta D$-$CH_4$ values consistent with previous studies. This alignment supports the representativeness of our inferred doubly substituted $CH_4$ isotopologues ratio source signatures for use alongside the bulk isotope ratios in global modelling of the $CH_4$ budget. Our database also provides further additional measurements of the bulk isotopes to aid in further work to refine the source signatures $\delta^{13}C$-$CH_4$ and $\delta D$-$CH_4$.

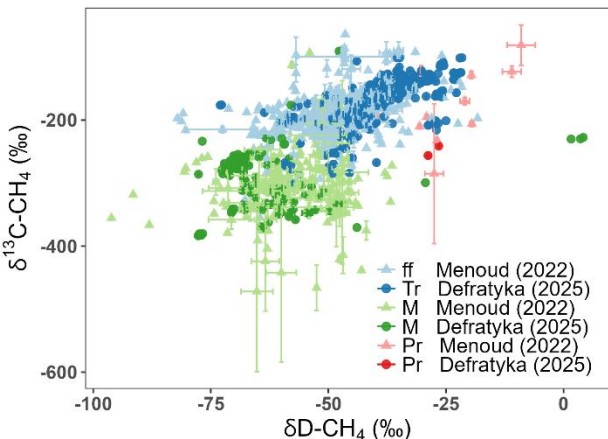

**Figure 1.** Database entries plotted as $\delta^{13}C$-$CH_4$ versus $\delta D$-$CH_4$ alongside the Menoud et al., 2022a database. Error bars are taken from original studies. ff: fossil fuels, Tr: thermogenic, M: microbial, Pr: pyrogenic.

**Table 1.** Comparison across the three databases of $\delta^{13}C$-$CH_4$ and $\delta D$-$CH_4$ by group type. The mean value is reported with ± 1 standard deviation, and minimum and maximum values in brackets.

| Group type | $\delta^{13}C$-$CH_4$ | | | $\delta D$-$CH_4$ | | |
|---|---|---|---|---|---|---|
| | samples | median (‰) | mean (‰) | samples | median (‰) | mean (‰) |

| | | | | | | |
|---|---|---|---|---|---|---|
| fossil fuels | | | -45.5 ± 9.1 | | | -185.5 ± 38.7 |
| Menoud et al. 2022 | 707 | -44.2 | [-82.1; -18.3] | 394 | -185.3 | [-355.0; -63.8] |
| fossil fuels | | | -44.9 ± 10.6 | | | -196.1 ± 48.6 |
| Sherwood et al. 2021 | 9477 | -43.0 | [-87.0; -14.8] | 3371 | -191.7 | [-415.0; -62.0] |
| thermogenic | | | -39.0 ± 9.6 | | | -169.2 ± 41.9 |
| Defratyka. et al. 2025 | 309 | -38.3 | [-73.0;-21.6] | 309 | -159.7 | [-300.2; -100.8] |
| microbial | | | -58.5 ± 8.5 | | | -309.7 ± 50.4 |
| Menoud et al. 2022 | 471 | -58 | [-96.1; -36.5] | 187 | -307.1 | [-472.0; -93.2] |
| microbial | | | -61.6 ± 6.9 | | | -304.0 ± 36.6 |
| Sherwood et al. 2021 | 131 | -62.4 | [-79.6; -45.5] | 20 | -304.0 | [-358.0; -205.0] |
| microbial | | | -62.9 ± 13.2 | | | -298.1 ± 47.7 |
| Defratyka. et al. 2025 | 120 | -66.8 | [-77.7; 4.2] | 120 | -294.7 | [-383.5; -90.5] |
| pyrogenic | | | -25.9 ± 7.7 | | | -176.7 ± 59.0 |
| Menoud et al. 2022 | 42 | -27.2 | [-42.7;-9.0] | 11 | -192.0 | [-285.0; -81.0] |
| pyrogenic | | | -26.0 ± 5.3 | | | -21.8 ± 15.5 |
| Sherwood et al. 2021 | 29 | -26.9 | [-33.4; -12.5] | 4 | -208.0 | [-232.0; -195.0] |
| pyrogenic | | | -27.7 ± 1.6 | | | -248.6 ±10.7 |
| Defratyka. et al. 2025 | 2 | -27.7 | [-28.8;-26.5] | 2 | -248.6 | [-256.1; -241.0] |

The references included in the database of doubly substituted $CH_4$ isotopologues comprise mostly peer-reviewed articles, with a smaller percentage from conference papers. The aggregated studies were carried out between 2014 and 2025 across 10 laboratories worldwide. As the aim of this study is to include all existing studies of doubly substituted isotopologue ratios, we also incorporated results from laboratory experiments, and of $CH_4$ dissolved in water (i.e. in oceans, wetlands, and inland waters), which were not included in bulk isotopes databases.

### 3.1.1. The structure of the database

For efficient utilization of the database, we start with parameters (column names) from the databases of Sherwood et al., (2017, 2021) and Menoud et al., (2022). Then, we added the parameters to better represent the characteristics of doubly substituted isotope ratio measurements. Selected parameters are described in the metadata of the database (https://dx.doi.org/10.5285/51ae627da5fb41b8a767ee6c653f83e6). Collection and analysis dates, along with instrument and measurement laboratory are included to facilitate comparison between studies. For each entry of $\Delta_{18}$, $\Delta^{13}CH_3D$ or $\Delta^{12}CH_2D_2$, the number of samples, measured value, uncertainty, and type of uncertainty are provided. The parameter "other tracers" was added to include information about other tracers collected alongside doubly substituted isotopologues and bulk isotope ratio measurements of $CH_4$. This parameter can be used to filter and group data for the further processing by database users. We also added the "lab field" parameter to make it easier to filter the database based on whether the sample was collected in the field or obtained from a laboratory experiment.

For samples collected from the field, we provided exact location (latitude and longitude), coming from the original article or approximate location, estimated based on geographical information in the article. The parameter "coordinates from primary source" was added to indicate if sampling location was given in the original article. We used the parameters documented by Menoud et al., (2022a) to describe the $CH_4$ source for field samples: group type, group, category and subcategory but with modifications to better reflect properties of $\Delta^{13}CH_3D$ and $\Delta^{12}CH_2D_2$ studies conducted so far (table 2). For example, in group type, we divided microbial sources into three categories: microbial terrestrial, microbial fossil fuels (microbial ff) and microbial marine. Additionally, we incorporated a parameter "sources specification" to add any information coming from the primary studies' publications that did not match the already included source parameters (e.g., thermodynamic disequilibrium or equilibrium, natural gas maturity, sources mixture). Parameters: "sample type", "reservoir type", "depth type" (i.e., unit of

285 reservoir depth from original paper) and "depth" were included for the description of field sampling conditions.

Whenever possible, we connected these groups and categories to the broadly used Selected Nomenclature for Air Pollution (SNAP) and Intergovernmental Panel on Climate Changes (IPCC, guidelines 2006) emissions categories for field samples (table 2). The Emissions Database for Global
Atmospheric Research (EDGAR) inventories are compatible with IPCC nomenclatures, which facilitates implementation of the database and comparison with existing emissions inventories (details in section 4.3.1). In the database, samples from laboratory experiments, ambient air, and volcano (both mud volcano and steam volcano) measurements are not linked to SNAP and IPCC categories. Also, the SNAPP and IPCC categories were not allocated to groundwater nor deep marine samples (i.e., marine
seeps, sediments, and pore fluid), as they represent insignificant sources of $CH_4$ to the atmosphere.

**Table 2.** Group type, group, category, and subcategory of $CH_4$ sources for field samples with SNAP and IPCC categories, based on source categories from Menoud et al. (2022).

| GROUP TYPE | GROUP | CATEGORY | SUB_CATEGORY | SNAP | IPCC 2006 |
|---|---|---|---|---|---|
| abiotic | exploitation | oil non-associated | natural gas | 5 | 1B2 |
| | | metal mine | groundwaters | - | - |
| | seeps | marine; temperate; volcanoes | hydrothermal vent, marine seep; hyperalkaline spring, hot spring, spring; mud volcano | - | - |
| ambient air | urban background | - | - | - | - |
| | mixture with $CH_4$ source | - | - | - | - |
| | clean background | - | - | - | - |
| microbial terrestrial | agriculture | rice paddies | rice paddies | 10 | 3C7 |
| | | ruminants | dairy cow | 10 | 3A1 |
| | exploitation | metal mine | groundwater | - | - |
| | seeps | temperate; volcanoes | groundwater, spring; mud volcano | - | - |
| | wetlands | polar (incl. boreal), temperate | lake, pond, swamp | 11 | 3B4 |
| microbial fossil fuel (microbial ff) | exploitation | coal | coal seam gas | 5 | 1B2 |
| | | biodegradation of oil, conventional | gas installation, natural gas, oil field | 5 | 1B2 |
| microbial marine | sediment | marine | marine sediment, pore fluid | - | - |
| | seeps | marine | cold seep, marine seep, pockmark | - | - |
| mixture | exploitation | conventional, unconventional, unconventional shale, oil non-associated, oil associated, coal | gas installation, natural gas, oil field, coal seam gas | 5 | 1B2 |
| | | metal mine | groundwater | - | - |
| | sediment | marine | marine sediment | - | - |
| | seeps | marine | marine seep | - | - |
| | | temperate | groundwater, hyperalkaline spring | - | - |
| | | volcanoes | mud volcano, steam volcano | - | - |
| | wetlands | polar (incl. boreal) | lake | 11 | 3B4 |
| others | exploitation | conventional | gas installation, natural gas | 5 | 1B2 |
| | | metal mine | groundwater | - | - |
| | sediment | marine | marine sediment | - | - |

| | | | | | |
|---|---|---|---|---|---|
| | seeps | temperate; volcanoes | groundwater; hydrothermal, steam vent, mud volcano, spring | - | - |
| | vehicle exhaust | - | - | | |
| | | | - | 7 | 1A3 |
| | wetlands | polar (incl. boreal) | lake | 11 | 3B4 |
| pyrogenic | fossil fuel burning, biomass burning | charcoal, oak logs | biomass burning | 11 | 3C1 |
| thermogenic | exploitation | conventional, unconventional, conventional oil associated, conventional oil non-associated, unconventional oil associated, unconventional oil non-associated, oil associated, oil non-associated, shale, unconventional shale | gas installation, natural gas | 5 | 1B2 |
| | sediment | marine; quartz-hosted inclusions | marine sediment; natural gas | - | - |
| | seeps | marine | hydrothermal vent, marine seep | - | - |
| | | volcanoes | hydrothermal, steam vent, mud volcano | - | - |
| | wetlands | polar (incl. boreal) | lake | 11 | 3B4 |

For samples coming from laboratory experiments, we added a specification of the type of laboratory experiment (e.g., abiotic or microbial methanogenesis, pyrolysis experiment, AOM or AeOM methanotrophy) in the group type column (table 3). Also, parameters "lab experiment type" and "lab experiment detail" were added to include details of conducted experiments. "Catalytic equilibration" experiments are focused on defining the thermal equilibration curve, used for the instruments calibration (Eldridge et al., 2019; Liu et al., 2024b; Ono et al., 2014; Wang et al., 2019; Young et al., 2017).

**Table 3.** Group type and laboratory experiment type for laboratory experiment samples

| GROUP TYPE | LAB EXPERIMENT TYPE |
|---|---|
| abiotic methanogenesis | high temperature abiotic |
| | low temperature abiotic |
| microbial methanogenesis | acetoclastic |
| | hydrogenotrophic |
| | methoxydotrophic |
| | methylotrophic |
| | methylphosphonate |
| pyrogenic methanogenesis | alkane pyrolysis |
| | coal pyrolysis |
| | ethane pyrolysis |
| | hydrous pyrolysis |
| | nonhydrous pyrolysis |
| | propane pyrolysis |
| | shale pyrolysis |
| AeOM methanotrophy | pure culture |
| AOM methanotrophy | enrichment culture |
| | field samples incubation |
| photochemical oxidation | control sample |
| | Cl oxidation |
| | OH oxidation |

| catalytic equilibration | bracketing calibration |
|---|---|
| mixing experiment | - |
| sediment incubation | aquatic environment sediment microbially enhanced coal bed CH$_4$ wetland soil |

Due to variations in measurement protocols across laboratories, uncertainties are reported in different ways and therefore we reported uncertainty per entry as described in the database. Most of the laboratories report one or two internal standard errors (int SE) to reflect precision based on HR-IRMS counting statistics (e.g., Ash et al., 2019; Douglas et al., 2016, 2017; Thiagarajan et al., 2020; Wang et al., 2023a; Young et al., 2017). Others use external reproducibility, expressed as one or two external standard deviations (ext SD) (Eldridge et al., 2019; Giunta et al., 2021; Wang et al., 2024a; Warr et al., 2021a). When one sample is measured more than once, one SE or two SE are reported as uncertainty in the database (Stolper et al., 2015; Wang et al., 2018; Wang et al., 2024). For some studies, uncertainty is reported as 95% confidence intervals (95% CI) (e.g., Beaudry et al., 2021; Lalk et al., 2024; Ono et al., 2014).

As studies were made over time by different laboratories, not all required database parameters were included in existing peer-reviewed articles. For the future, proposed parameters should ideally be published with data. Additionally, a consistent description of CH$_4$ sources, group type, group, category, subcategory and laboratory experiment type, using the parameters proposed in table 2 and 3 is encouraged to facilitate interpretation and intercomparison between laboratories and methods.

## 4. Results and discussion

### 4.1. Data summary

Out of all data entries, field samples comprise 958 entries, while 517 entries come from laboratory experiments. Of these, 53% of entries report only $\Delta_{18}$ or $\Delta^{13}CH_3D$. Potentially, the lack of $^{12}CH_2D_2$ measurements can hinder data interpretation, especially for microbial, abiotic or mixed samples, where $\Delta^{13}CH_3D$ and $\Delta^{12}CH_2D_2$ can be modified differently (e.g., Douglas et al., 2017; Giunta et al., 2019; Gruen et al., 2018; Thiagarajan et al., 2020; Warr et al., 2021; Young et al., 2016, 2017). To avoid data misinterpretation, other tracers, for example radiocarbon or seismic reflection data, must be measured alongside to $\Delta^{13}CH_3D$ (e.g., Chowdhury et al., 2024; Douglas et al., 2020).

Regarding the parameter "group type", thermogenic samples contribute 32% to the field samples, while there is low representation of pyrogenic samples (0.21% of field samples) (figure 2). "Others" is a broad group type of field samples with ambiguous origin from various sources (e.g., natural gas, groundwaters from metal mines, marine and mud volcano samples), where it was not possible to clearly determine the group type based on isotopes and other tracers. Hypothesized origins of these samples are given as 'source specification' parameter in the database. Also, vehicle exhaust samples are classified as "others", as different processes can cause CH$_4$ emissions from the exhaust (Sun et al., 2025b). Additionally, for samples where two different sources of CH$_4$ were mixed, indicated as group type 'mixture', more information on the type of mixture is added under the parameter "source specification" in the database. For ambient air "group type", distinction between background samples and mixture of ambient air and gas coming from CH$_4$ source (e.g., gas sample collected above wetland, (Haghnegahdar et al., 2024; Sun et al., 2025b)) was made using the "group" parameter.

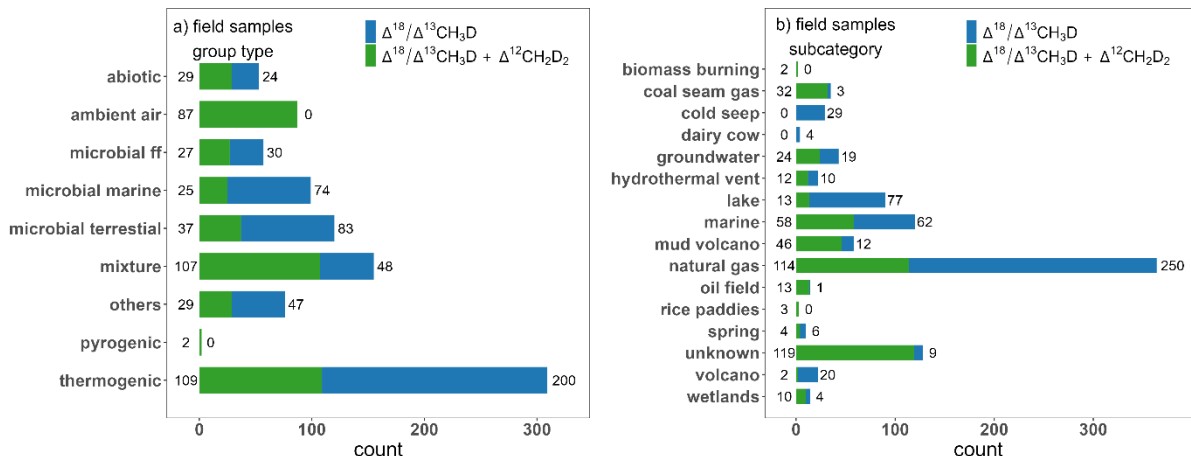

**Figure 2.** Frequency of entries for field samples categorised by a) group type and b) subcategories.

The distribution of measurements in $\Delta^{13}CH_3D$ versus $\Delta^{12}CH_2D_2$ space is presented in figure 3, both for field samples and laboratory experiments. To simplify data interpretation, field samples categorized as "others" or "mixture" are omitted. Also, samples where ambient air is mixed with the gas from $CH_4$ source are omitted. The majority of thermogenic samples fall close to the thermodynamic isotopic equilibrium curve, with a few samples having more depleted $\Delta^{12}CH_2D_2$ than predicted (details in section 4.2.). Microbial marine and microbial ff samples are near or at thermodynamic isotopic equilibrium but with some enrichment relative to equilibrium observed. Most of the microbial terrestrial samples (e.g., lakes, wetlands or agriculture) are clearly depleted in both $\Delta^{13}CH_3D$ and $\Delta^{12}CH_2D_2$, relative to the equilibrium. Different ratios for microbial terrestrial compared to microbial ff and microbial marine suggests different methanogenesis reactions or additional processes, such as methanotrophy or mixed patterns of microbial carbon cycling within in these environments (details in section 4.2.). Regarding abiotic $CH_4$, most of the samples are out of thermodynamic isotopic equilibrium (e.g., Douglas et al., 2020; Labidi et al., 2020; Lin et al., 2023; Young et al., 2017). It must be noted, that abiotic $CH_4$ is empirically one of the least well characterized endmembers, both in terms of field and laboratory studies.

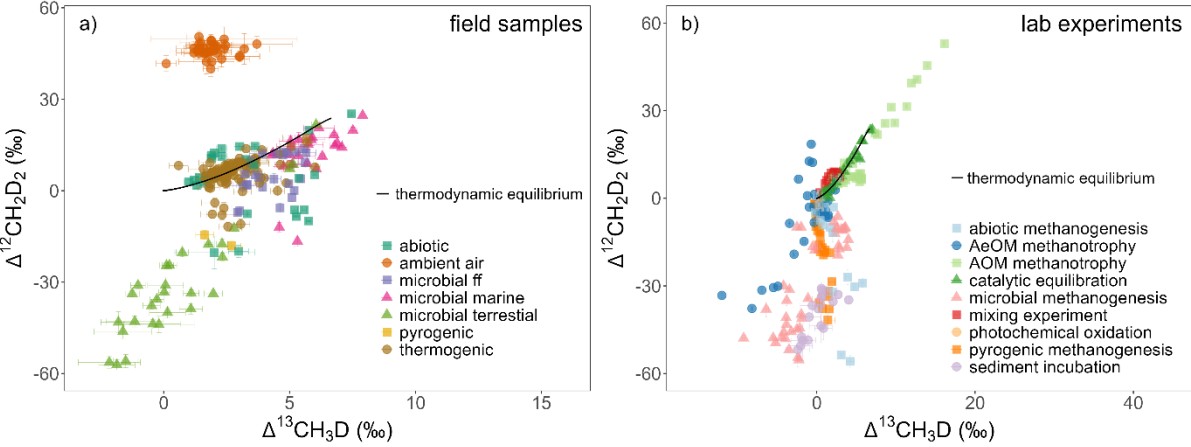

**Figure 3.** Database entries plotted as $\Delta^{13}CH_3D$ versus $\Delta^{12}CH_2D_2$. Error bars are taken from original studies (details in section 3.1.1). a) fields samples based on 247 entries, where samples categorized as "others", "mixture" and "ambient air mixed with $CH_4$ source" are omitted for simplicity. b) laboratory experiments based on 210 entries. Laboratory experiments with deuterium-enriched water substrate (Gruen et al., 2018; Li et al., 2024, 2025a; Taenzer et al., 2020) are not included as they do not appear under normal incubation or environmental conditions. A solid black line represents the thermodynamic isotopic equilibrium curve, using equations from Young et al., (2017).

For laboratory experiments, the deviation from thermodynamic isotopic equilibrium depends on the studied methanogenesis pathway or the type of methanotrophy (aerobic (AeOM) versus anaerobic (AOM) $CH_4$ oxidation) (details in section 4.2.). For example, AOM methanotrophy experiments show a large enrichment for both $\Delta^{13}CH_3D$ and $\Delta^{12}CH_2D_2$ (Liu et al., 2023; Ono et al., 2021). Notably, Gruen et al., (2018), Li et al., (2024, 2025a), and Taenzer et al., (2020), carried out incubations with deuterium-enriched substrate to explore mechanisms behind combinatorial effects. Thus, observed clumped isotopologues do not represent the isotopic values of natural-occurring microbial $CH_4$ and should be carefully re-interpreted.

Regarding pyrogenic methanogenesis, some samples have doubly-substituted isotope ratio compositions consistent with thermodynamic isotopic equilibrium, while others create more depleted values, due to a combination of kinetic isotope effects, combinatorial effects, and varying degrees of hydrogen isotope exchange (Dong et al., 2021; Eldridge et al., 2023; Shuai et al., 2018a). The abiotic synthesis of $CH_4$ in laboratory-controlled experiments shows enriched $\Delta^{13}CH_3D$, consistent with thermodynamic isotopic equilibrium, associated with systematically depleted $\Delta^{12}CH_2D_2$, due to combinatorial effects (Young et al., 2017, Labidi et al., 2024).

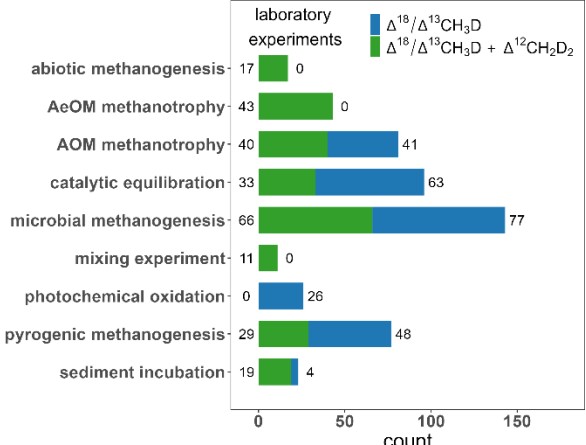

**Figure 4.** Frequency of entries for laboratory experiments categorised by group type

About 27% of the laboratory experimental entries come from studies on microbial methanogenesis, focused on various pure cultures of methanogenic archaea (e.g., acetoclastic, hydrogenotrophic and methylotrophic methanogenesis) (figure 4) (Douglas et al., 2016, 2020; Giunta et al., 2019; Gruen et al., 2018; Rhim and Ono, 2022; Stolper et al., 2015; Warr et al., 2021a; Young et al., 2017). Notably, Li et al. (2025a) conducted methanogenesis experiment where few data points come from extremely deuterium-enriched water (δD of water about 3000 ‰ and 8000 ‰). Such high $\delta^2H$ of water cannot be found in the nature, thus obtained $CH_4$ has very atypical isotopic values (figure 5).

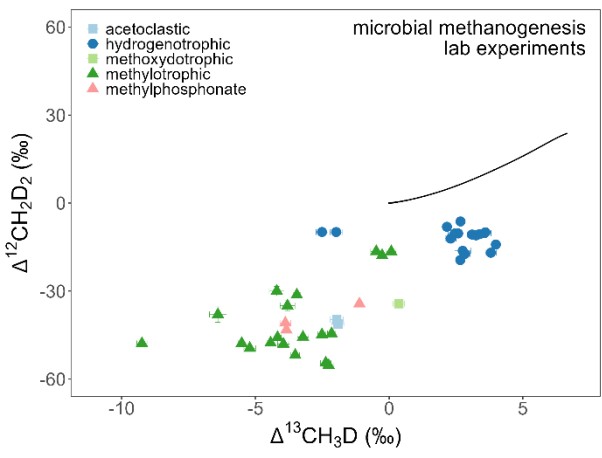

**Figure 5.** $\Delta^{13}CH_3D$ versus $\Delta^{12}CH_2D_2$ for microbial methanogenesis laboratory experiments. Laboratory experiments with deuterium-enriched water substrate (Gruen et al., 2018; Li et al., 2024, 2025a; Taenzer et al., 2020) are not included as they do not appear under normal incubation or environmental conditions.

4.4% of laboratory samples, classified as sediment incubation, were collected in the field and incubated in controlled laboratory conditions (Douglas et al., 2020; Haghnegahdar et al., 2023, 2024; Ijiri et al., 2018; Wang et al., 2024a). A single laboratory experiment focused on photochemical oxidation by OH and Cl was also conducted, however, only $\Delta^{13}CH_3D$ was measured (Whitehill et al., 2017). A laboratory experiment focused on mixing of two $CH_4$ sources, containing different bulk isotopic compositions, was conducted to confirm mixing curve delivered from theoretical calculation, related to the definition of $\Delta^{13}CH_3D$ and $\Delta^{12}CH_2D_2$ (Young et al., 2016).

### 4.2. State of knowledge about $CH_4$ doubly-substituted isotopologue ratios

Methane is produced at the surface and in subsurface environments via biogenic (microbial), thermogenic, or abiotic processes, while the majority of the $CH_4$ emitted to the atmosphere comes from microbial, thermogenic, and pyrolytic (biomass and biofuel burning) sources (e.g., Saunois et al., 2025; Schoell, 1988; Stolper et al., 2018). Thermogenic $CH_4$ forms by the thermally-activated breakdown of organic molecules, where 'primary thermogenic' is a term used to describe $CH_4$ produced from kerogen and 'secondary thermogenic' is used to describe the breakdown of long-chain hydrocarbons (e.g., Lalk et al., 2023; Stolper et al., 2018). Stolper et al. (2014b) proposed that thermogenic $CH_4$ is predominantly in thermodynamic isotopic equilibrium at its formation temperature, which was supported by studies focused on natural gas or volcanic samples (Beaudry et al., 2021; Douglas et al., 2016, 2017; Jiang et al., 2024; Kim et al., 2023; Rumble et al., 2018; Stolper et al., 2014b, 2015, 2018; Thiagarajan et al., 2020; Wang et al., 2015; Xie et al., 2021; Young et al., 2017). Formation temperatures calculated from doubly substituted isotope ratio measurements can help to determine the natural gas maturity and distinguish "atypical" thermogenic gas (from shallow or immature systems to deep or over-mature systems) from abiotic $CH_4$ (Jiang et al., 2024; Kim et al., 2023; Li et al., 2025b; Shuai et al., 2018b). Some exceptions of doubly substituted isotope ratios deviating from thermodynamic isotopic equilibrium were observed from unconventional, oil-non-associated or oil-associated gas reservoirs (figure 6) (Douglas et al., 2017; Kim et al., 2023; Lalk et al., 2022; Stolper et al., 2018; Xie et al., 2021), which is consistent with laboratory pyrolysis experiments and gas generation models implying at least partly kinetically-driven signatures (Dong et al., 2021; Eldridge et al., 2023; Shuai et al., 2018a; Xia and Gao, 2019). For low maturity or oil-associated natural gas, a contribution from microbial sources can occur, for example due to $CH_4$ generation during oil biodegradation (e.g., secondary microbial $CH_4$). However, the likelihood that microbial $CH_4$ has both $\Delta^{13}CH_3D$ and $\Delta^{12}CH_2D_2$ within the thermogenic range remains low (Giunta et al., 2019; Lalk et al., 2022; Thiagarajan et al., 2020; Xie et al., 2021).

Microbial $CH_4$ is produced by microorganisms via three main pathways: hydrogenotrophic, acetoclastic, and methylotrophic methanogenesis, with the first two being the predominant (Conrad, 2005; Thauer, 1998). Typically, subsurface microbial $CH_4$ from geological basins is mostly generated through the hydrogenotrophic pathway, where doubly substituted isotope ratios tend towards thermodynamic isotopic equilibrium (figure 3 and 5) (Ash et al., 2019; Douglas et al., 2016, 2017, 2020; Giunta et al., 2019; Shuai et al., 2021; Stolper et al., 2015; Thiagarajan et al., 2020; Wang et al., 2024a; Warr et al., 2021a; Young et al., 2017). Studies of pore water from the Michigan Basin, showed that deep subsurface $CH_4$ can also be generated by acetoclastic methanogenesis at thermodynamic isotopic equilibrium for $^{13}CH_3D$ but at substantial disequilibrium for $^{12}CH_2D_2$ (Jautzy et al., 2021). The majority of microbial $CH_4$ from shallow freshwater environments is generated during acetoclastic methanogenesis, which can result in strong depletion for both $^{13}CH_3D$ and $^{12}CH_2D_2$ (figure 3 and 5)

(Conrad, 2005; Douglas et al., 2016, 2017, 2020; Haghnegahdar et al., 2024; Li et al., 2025a; Stolper et al., 2014b; Wang et al., 2015; Whiticar, 1999; Young et al., 2017). In systems with presumed slow $CH_4$ generation rates, favouring enzymatic isotopic reversibility, microbial $CH_4$ likely can form at or near thermodynamic isotopic equilibrium, while in systems with rapid $CH_4$ formations, microbial $CH_4$ tends to depart from thermodynamic isotopic equilibrium (Douglas et al., 2020; Shuai et al., 2021; Stolper et al., 2015; Wang et al., 2015).

Methane can also be produced abiotically, for example via Sabatier reactions linked to hydrogen production from serpentinization in hydrothermal systems (Cumming et al., 2019; Douglas et al., 2017; Labidi et al., 2020; Nothaft et al., 2021; Ojeda et al., 2023; Suda et al., 2022; Wang et al., 2018; Young et al., 2017). It has been observed from deep groundwater seeps accessed via or within deep subsurfaces layers, for instance in metal mines, where it can also mix with microbial $CH_4$ followed by re-equilibration (Nothaft et al., 2021; Warr et al., 2021a; Young et al., 2017). Typically, abiotic $CH_4$ is produced at temperatures exceeding 250 °C in seafloor hydrothermal fluids or in the continental seeps, springs and fracture waters at temperatures lower than 100 °C (Etiope and Sherwood Lollar, 2013; Labidi et al., 2024; Young et al., 2017). During controlled laboratory synthesis under hydrothermal conditions, the majority of the $\Delta^{13}CH_3D$ measurements closely reflect the temperature of abiotic $CH_4$ generation (based on thermodynamic isotopic equilibrium). $\Delta^{12}CH_2D_2$ was observed with depletions down to -40‰, which can be attributed to a D/H combinatorial effect associated with the various steps of hydrogen addition to carbon occurring during $CH_4$ formation (Labidi et al., 2024).

Using doubly substituted isotope ratio measurements, the mixed thermogenic-microbial origin of $CH_4$ was observed in marine environments, including $CH_4$ clathrates (Giunta et al., 2021; Zhang et al., 2021), lakes (Douglas et al., 2016), mud volcanoes (Lin et al., 2023; Liu et al., 2024a; Rumble et al., 2018), oil fields (Tyne et al., 2021) and natural gas (Douglas et al., 2017; Giunta et al., 2019; Kim et al., 2023; Lalk et al., 2022; Stolper et al., 2014b, 2015; Thiagarajan et al., 2020, 2022). Mixing between different $CH_4$ sources (containing different bulk isotopic compositions) in different proportions creates a non-linear relationship in $\Delta^{12}CH_2D_2$ vs $\Delta^{13}CH_3D$ space. Measurement of both doubly-substituted isotope ratios therefore provides additional information to help define the mixed end members and understand if physical or chemical transformation processes have taken place (e.g., Douglas et al., 2016; Young et al., 2016; Zhang et al., 2021).

Notably, existing studies showed a range of doubly-substituted isotope ratios for mud volcano samples, suggesting their different origins (thermogenic, microbial, abiotic or mixed) and potentially reflecting subsequent alteration processes such as AOM (Ijiri et al., 2018; Lalk et al., 2022; Lin et al., 2023; Liu et al., 2023, 2024a; Rumble et al., 2018). Additionally, $\Delta^{13}CH_3D$ was used to demonstrate a microbial origin of $CH_4$ in deep subsurface coal beds in the north-western Pacific (Inagaki et al., 2015) and shallow subsurface mud volcano in the Nankai accretionary complex (Ijiri et al., 2018), which could otherwise be incorrectly identified as thermogenic sources. Also, $\Delta^{12}CH_2D_2$ vs $\Delta^{13}CH_3D$ suggested mixing of thermogenic and microbial $CH_4$ in coal bed reservoirs (Wang et al., 2024b, c).

Combinatorial effects occur when a molecule contains indistinguishable atoms of the same element derived from pools with different isotope ratios. This purely mathematical phenomenon comes from the definition of doubly-substituted isotope ratio in reference to the stochastic distribution and has been predicted theoretically (Röckmann et al., 2016; Yeung, 2016) and demonstrated experimentally for $CH_4$ (Labidi et al., 2024; Taenzer et al., 2020; Wang et al., 2024a). Among the two mass-18 isotopologues of $CH_4$, only $\Delta^{12}CH_2D_2$ can be influenced by combinatorial effects, as it features two indistinguishable deuterium substitutions for hydrogen. Combinatorial effects for $\Delta^{12}CH_2D_2$ values must be taken into account in low-temperature abiotic or biotic systems where the hydrogen atoms of $CH_4$ originates from multiple reservoirs, which has been observed in microbial samples (Giunta et al., 2019; Jautzy et al., 2021; Young et al., 2017), mud volcanos (Liu et al., 2024a), natural gas (Shuai et al.,

2021; Xie et al., 2021), or during abiotic, microbial and pyrogenic methanogenesis experiments (Dong et al., 2021; Eldridge et al., 2023; Labidi et al., 2024; Li et al., 2025a). Notably, Eldridge et al., (2023) showed that combinatorial effects alone cannot explain the non-equilibrium of $\Delta^{12}CH_2D_2$, observed in their pyrogenic methanogenesis experiments focused on $CH_4$ formation from methyl precursors (i.e. ethane). They pointed out the role of other important processes such as the influence of kinetic isotope effects and inheritance reactions (i.e., inheriting 'clumps' from methyl groups in the precursor molecule), in addition to combinatorial effects.

Before emission to the atmosphere, $CH_4$ can be consumed through aerobic oxidation (AeOM) or anaerobic oxidation (AOM). In terrestrial ecosystems (e.g., wetlands) and oxygenated marine water columns, AeOM plays a crucial role, while in gas seeps and sulphate-rich marine sediments, AOM likely dominate causing inhibition of $CH_4$ emissions to the atmosphere (e.g., Wang et al., 2016 and references therein). Minor depletions in $\Delta^{13}CH_3D$ and $\Delta^{12}CH_2D_2$ were observed in AeOM-dominated systems, but low-temperature equilibrium or significant enrichments in $\Delta^{13}CH_3D$ and $\Delta^{12}CH_2D_2$ were observed in the case of AOM (figures 3 and 7) (Giunta et al., 2022; Kim et al., 2023; Liu et al., 2023; Ono et al., 2021). One hypothesis states that the reversibility of initial steps of AOM promotes thermodynamic equilibration (Ash et al., 2019; Giunta et al., 2022; Ono et al., 2021; Zhang et al., 2021). Alternatively, another hypothesis proposes that near-thermodynamic equilibrium of doubly substituted isotope ratios in marine sediments can be attained via a slow rate of methanogenesis, with reversible enzymatic reaction steps (Douglas et al., 2020; Shuai et al., 2021; Stolper et al., 2015; Wang et al., 2015). As AeOM and AOM have distinctive kinetic isotope effects in natural settings, doubly-substituted isotope ratios may be used to track and differentiate both AeOM and AOM in nature (Adnew et al., 2025; Ash et al., 2019; Giunta et al., 2019, 2022; Krause et al., 2022; Li et al., 2024; Tyne et al., 2021; Warr et al., 2021b; Zhang et al., 2021).

In the troposphere, reaction with OH is the primary removal mechanism of $CH_4$ (90%), with other minor contributions from microbial oxidation in soils and vegetation, loss to the stratosphere, and reactions with tropospheric Cl (e.g., Saunois et al., 2025). Overall, isotopologues containing bonds of lighter isotopes are preferentially removed through photochemical oxidation, leading to an enrichment in heavier isotopologues of the remaining $CH_4$ pool (table S2) (e.g., Haghnegahdar et al., 2017; Whitehill et al., 2017). Laboratory experiments showed that photochemical oxidation by OH has only a minor impact on $\Delta^{13}CH_3D$ of tropospheric $CH_4$ (i.e. the $^{13}$C-D bond does not react significantly slower than that calculated based on equivalent singly substituted reactants) (Whitehill et al., 2017). Thus, measurements of $\Delta^{13}CH_3D$ in the atmosphere can provide constraints on $CH_4$ source strengths, while $\Delta^{12}CH_2D_2$ is predicted to provide information on $CH_4$ sink strength, as implemented in global scale atmospheric models (Chung and Arnold, 2021; Haghnegahdar et al., 2017; Whitehill et al., 2017). Aside from the atmospheric models, Wang et al. (2023b) used machine learning incorporated with a random forest model to predict steady-state atmospheric $CH_4$ doubly substituted isotope ratios. The first measurements of the doubly substituted isotope ratio of $CH_4$ in the atmosphere were more depleted for both $\Delta^{13}CH_3D$ and $\Delta^{12}CH_2D_2$ than predicted by atmospheric models and available source signature information (Chung and Arnold, 2021; Haghnegahdar et al., 2017, 2023; Sivan et al., 2024). Haghnegahdar et al. (2023) proposed that differences between measurements and predictions required depleted doubly substituted isotopic signature values for the (total) source flux than previously assumed. On the other hand, Sivan et al. (2024) highlighted that the observed discrepancy could also be caused by inaccuracy in the theoretical values of the kinetic isotopic effect (KIE) of $CH_4$ reactions with OH, Cl and soils sinks. They indicated that a small adjustment in the sink KIE, along with slightly lower source mixture than previously assumed, could align atmospheric and source doubly substituted isotopic signatures (Sivan et al., 2024).

### 4.3. Data representatives and importance for atmospheric sciences

The distribution of $\Delta^{13}CH_3D$ and $\Delta^{12}CH_2D_2$ derived from field samples per simplified subcategory is plotted on figure 2b, while figures 6 and 7 present box plots for measured doubly substituted isotopes

from field samples and laboratory experiments, respectively. For simplicity, in figures 2b and 6, and thereafter, some subcategories are merged. Gas installation and natural gas subcategories are merged into natural gas. Hot spring, spring, and hyperalkaline spring are unified as spring. Marine sediment, marine seep, pore fluid and pockmark are grouped as a marine subcategory. Hydrothermal and volcano steam samples are unified as volcano. Finally, swamp and ponds are merged as wetlands, while lakes are in a separate subcategory. Around 40% of field samples were collected from reservoirs of natural gas (figure 2b). About 3% of field samples come from coal seam gas and 12.5% come from microbial terrestrial sources. There is a significant representation of marine (12.5% of field samples) and volcano mud samples (6% of field samples), although, their emissions to the atmosphere are negligible. For samples categorized as microbial terrestrial, the majority of entries come from lakes (75% of microbial terrestrial), with a small contribution from agriculture (6%) or wetland (12%) samples, which are significant $CH_4$ emitters to the atmosphere. Only $\Delta^{13}CH_3D$ was measured for four ruminants samples (Lopes et al., 2016; Wang et al., 2015). Only three samples from rice paddies have so far been collected, where both $\Delta^{13}CH_3D$ and $\Delta^{12}CH_2D_2$ were measured (Haghnegahdar et al., 2023; Wang et al., 2023a). So far, no waste samples have been collected directly from the source for studies of doubly substituted isotope ratios. The recent studies of Sun et al. (2025a) focused on collection of big volume ambient air samples, where background air was mixed with gas coming from microbial $CH_4$ sources, like wetlands and landfills. Application of a Keeling plot method (Pataki, 2003), allowed determination of targeted sources (Sun et al., 2025a).

Published $\Delta^{13}CH_3D$ and $\Delta^{12}CH_2D_2$ for natural gas are consistent with a thermogenic origin (figure 3 and 6, table S3 and S4). Observed outliers come from low maturity or oil-associated natural gas where a microbial contribution could be significant (Kim et al., 2023; Lalk et al., 2022; Thiagarajan et al., 2020; Xie et al., 2021). No significant variation has been observed in measurements made of biomass burning, dairy cows (ruminants), or rice paddies within the available, limited dataset but this may not reflect the variation within the true population (table S3 and S4). Significant variation in both $\Delta^{13}CH_3D$ and $\Delta^{12}CH_2D_2$ is observed for spring and mud volcano subcategories, as these samples have varying microbial, thermogenic, abiotic, or mixed origins. Finally, a wide distribution is observed for lake samples, potentially originating from seasonal variation in $CH_4$ production, oxidation in the lake subsurface or methanogenic metabolisms involved (Lalk et al., 2024).

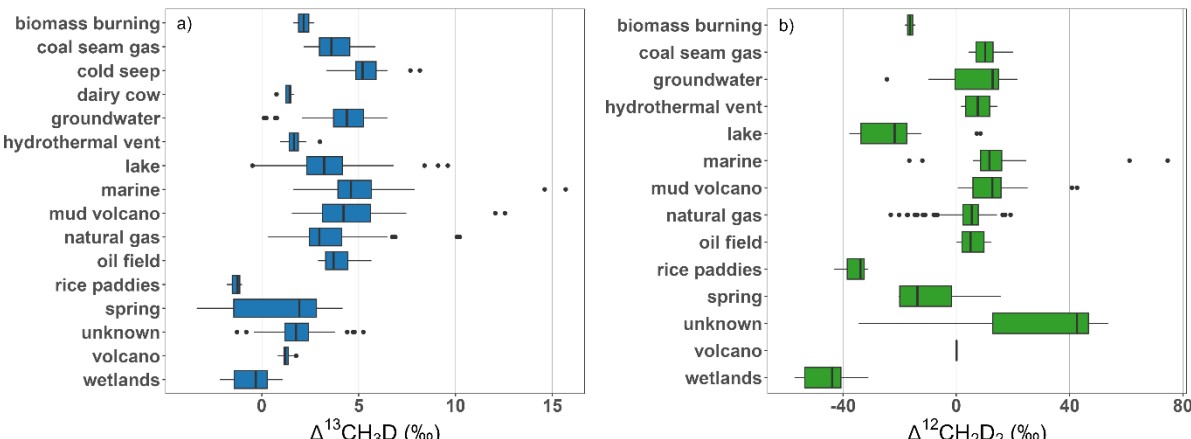

**Figure 6.** Summary of the distribution of measurement results, a) $\Delta^{13}CH_3D$ and b) $\Delta^{12}CH_2D_2$ from field studies based on simplified subcategories as described in section 4.3.

For the laboratory experiments, culturing of different strains of archaea and wide variations in experimental parameters resulted in a wide distribution of observed doubly substituted isotopic compositions, especially for $\Delta^{12}CH_2D_2$ (figure 7, table S5 and S6). AOM methanotrophy experiments show significant enrichment in both $\Delta^{13}CH_3D$ and $\Delta^{12}CH_2D_2$ relative to the other categories.

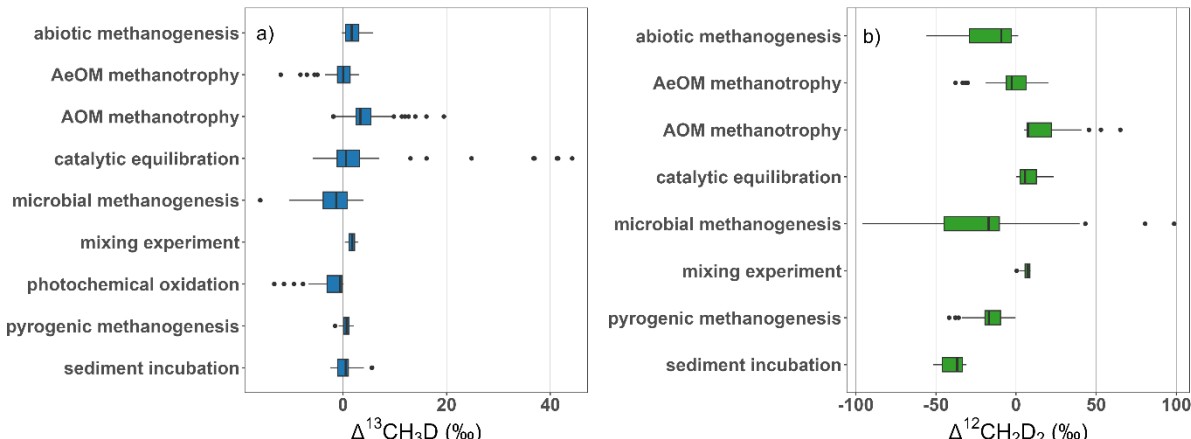

**Figure 7.** Summary of the distribution of measurement results, a) $\Delta^{13}CH_3D$ and b) $\Delta^{12}CH_2D_2$ from laboratory studies based on group types as described in section 4.1. The outliers for catalytic equilibration come from the sample measured at the beginning of the experiment, when equilibration on the catalyst did not start yet.

### 4.3.1. Evaluation of the database in relation to emission to the atmosphere

On a global scale, using a bottom-up approach (e.g., using data-driven and process based models for natural sources and inventories for anthropogenic sources) for the year 2020, anthropogenic emissions contribute about 54% of the $CH_4$ emissions to the atmosphere, originating from agriculture (40%), fossil fuel extraction and use (34%), waste (19%) and anthropogenic biomass burning (7%) (Saunois et al., 2025). Wetlands account for most of the natural $CH_4$ emissions (51%), with a significant contribution from inland freshwaters (35%) and remaining emission coming from other sources, including onshore and offshore geological emissions (e.g., mud volcanoes, volcanoes, vents, seepages) (Saunois et al., 2025). Regarding the main $CH_4$ emitters to the atmosphere, natural gas and oil are the most represented emission category in the doubly substituted $CH_4$ isotopologue database (39% of field samples), while coal seams gas samples represent 4% of the field samples in the database. There are no reported measurements of $\Delta^{12}CH_2D_2$ for ruminants (4 samples for $\Delta^{13}CH_3D$ values), and no records of either $\Delta^{13}CH_3D$ or $\Delta^{12}CH_2D_2$ from directly sampled waste. Additionally, there is a very limited sample size for some important emissions subcategories such as biomass burning (0.2%) and rice paddies (0.3%). As field sampling is time consuming and location-constrained, measurements made this far do not reflect a realistic spatio-temporal variation of doubly substituted isotope ratios, both for anthropogenic and natural $CH_4$ sources. With such limited studies, the current estimated $\Delta^{13}CH_3D$ and $\Delta^{12}CH_2D_2$ source signatures may not be representative. Thus, some assumptions on the source signature inputs to global scale models of double subsisted isotope ratios have to be made (table 4). To better reflect $\Delta^{13}CH_3D$ and $\Delta^{12}CH_2D_2$ of $CH_4$ emission sectors, further sampling should be focused on underrepresented $CH_4$ sources and on numerous conditions affecting emissions from individual sectors, for example impact of reservoir depth and coal type for coal seam gas or impact of diet and living conditions for rumen (table 4). An effort should be made to measure $\Delta^{13}CH_3D$ and $\Delta^{12}CH_2D_2$ from thawing permafrost, as it may be a significant source of $CH_4$ to the atmosphere in the future (Douglas et al., 2020; Ellenbogen et al., 2024; Walter Anthony et al., 2024).

**Table 4.** Global $CH_4$ emissions and inferred doubly substituted $CH_4$ isotope ratio signatures with remarks on the current representativeness of main $CH_4$ sources to the atmosphere and requirements for future studies. Uncertainties of global emissions are reported as [min-max] range.

| Group | Category | Global flux | $\Delta^{13}CH_3D$ [‰] | | | $\Delta^{12}CH_2D_2$ [‰] | | | Remarks | |
| | | | Average signature | Range | Samples number | Average signature | Range | Samples number | Representativeness | Existing models assumptions |
|---|---|---|---|---|---|---|---|---|---|---|

| | | [Tg CH$_4$ yr$^{-1}$] Bottom-up [1] | | | | | | | | |
|---|---|---|---|---|---|---|---|---|---|---|
| fossil fuels | coal seam gas | 41 [38-43] | 3.77 | 2.16; 5.87 | 35 | 10.20 | 4.25; 20.05 | 32 | coal samples collected for sediment incubation experiments; no samples from mine ventilation; no information about impact of depth of coal seams or type of coal extraction | Whitehill et al. (2017): only Δ$^{13}$CH$_3$D, a common signature for lakes, landfill, all fossil fuels and biomass burning, estimated based on Wang et al. (2015); Haghnegahdar et al (2017): assumed a common signature for all fossil fuels and biomass burning; Chung and Arnold (2021): Δ$^{12}$CH$_2$D$_2$ as in Haghnegahdar et al (2017), Δ$^{13}$CH$_3$D different to Haghnegahdar et al (2017) but common to all fossil categories; Haghnegahdar et al (2023): a common signature for all fossil fuels |
| | natural gas | 74 [67-80] [2] | 3.36 | 0.30; 10.22 | 381 | 3.79 | -23.13; 19.15 | 114 | Emission from natural gas and oil merged in models and inventories; the best representation in the database; samples taken from sources with or without thermodynamic equilibrium; samples taken from different extraction regions; future sampling should be focused on underrepresented regions and various oil and gas infrastructure | |
| | oil field | | 3.98 | 2.90; 5.66 | 14 | 6.13 | 0.01; 12.46 | 13 | | |
| microbial (except microbial fossil fuels) | dairy cow | 117 [114-124] [3] | 1.32 | 0.76; 1.66 | 4 | N/A | N/A | N/A | only Δ$^{13}$CH$_3$D measured; uncertain if dairy cow isotope ratio is representative for all ruminants and manure; critical demand of more sampling (type of rumen, impact of diet and living conditions, regional variation, different manure management systems), demand for Δ$^{12}$CH$_2$D$_2$ measurements | Whitehill et al. (2017): only Δ$^{13}$CH$_3$D, a common signature for ruminants and rice paddies, estimated based on Wang et al. (2015); Haghnegahdar et al (2017): different signature using three different scenarios; Chung and Arnold (2021): Δ$^{12}$CH$_2$D$_2$ as in Haghnegahdar et al. (2017), Δ$^{13}$CH$_3$D based on cow rumen measurements. Haghnegahdar et al (2023): signatures based on interpretation of their wetland measurements. |
| | lake | 112 [49-202] [4] | 3.35 | -0.48; 9.60 | 91 | -20.97 | -37.76; 8.55 | 13 | Samples taken mostly from lakes in the US with some | Chung and Arnold (2021): Δ$^{12}$CH$_2$D$_2$ as in Haghnegahdar et al |

| | | | | | | | | | |
|---|---|---|---|---|---|---|---|---|---|
| | | | | | | | | contribution from European and Chinese lakes; only one study focused on seasonal variation, but no $\Delta^{12}CH_2D_2$ measurement (Lalk et al. 2024) | (2017), $\Delta^{13}CH_3D$ based on freshwater measurements |
| | rice paddies | 32 [29-37] | -1.36 | -1.80; -1.02 | 3 | -36.04 | -43.17; -31.11 | 3 | Three field samples over two studies (two samples from China and one from the US), demand of increased spatial representation and samples from different rice paddies management systems (e.g., flooding, soil, rice variety) | Chung and Arnold (2021): $\Delta^{12}CH_2D_2$ as in Haghnegahdar et al (2017), $\Delta^{13}CH_3D$ based on cow rumen measurements. Haghnegahdar et al (2023): signatures based on interpretation of their wetland measurements |
| | waste[6] | 71 [60-84] | -1.3 | N/A | N/A | -38.8 | N/A | N/A | One of the main sources of $CH_4$ to the atmosphere; no representation of direct samples in the database; one study of mixed ambient air and landfill air (Sun et al. 20125), critical demand of samples from solid landfill, wastewater treatment and biogas, including sampling in different regions and seasons | Haghnegahdar et al. (2017): different signature using three different scenarios; Chung and Arnold (2021): $\Delta^{12}CH_2D_2$ as in Haghnegahdar et al (2017), $\Delta^{13}CH_3D$ based on cow rumen measurements. Haghnegahdar et al (2023): signatures based on interpretation of their wetland measurements |
| | wetlands | 161 [131-198] | -0.49 | -2.16; 1.08 | 14 | -45.61 | -57.16; -31.02 | 10 | Samples taken only from wetlands in the US; demand for samples from different wetland regions, including tropical (significant $CH_4$ emitter) and polar wetlands and permafrost | Haghnegahdar et al (2017): category divided into boreal and tropical wetlands. Chung and Arnold (2021): $\Delta^{12}CH_2D_2$ as in Haghnegahdar et al (2017), $\Delta^{13}CH_3D$ based on freshwater measurements |
| pyrogenic | biomass burning | 27 [20-41] [5] | 2.16 | 1.63; 2.69 | 2 | -16.31 | -18.12; -14.49 | 2 | Demand for samples from different type of biomass and biofuel; need for examination of the impact of burning conditions on isotope ratios (few laboratory | Haghnegahdar et al (2017): assumed thermodynamic equilibrium, common signature for all fossil fuels and biomass burning; |

|  |  |  |  |  |  |  |  | experiments conducted) |  |

1) CH$_4$ global flux from Saunois et al. 2025 for the year 2020
2) CH$_4$ global flux for natural gas and oil merged into one category in Saunois et al. 2025
3) enteric fermentation & manure category in Saunois et al. 2025
4) inland freshwater category in Saunois et al. 2025
5) biomass and biofuel burning together from Saunois et al. 2025
6) $\Delta^{13}CH_3D$ and $\Delta^{12}CH_2D_2$ of waste sector from indirect measurement (e.g., ambient air mixed with gas from landfill) from Sun et al. 2025

In addition to increasing the sampling frequency for the main CH$_4$ sources, an effort should also be made to extend sampling to other areas with significant CH$_4$ emissions to the atmosphere, including super-emitters. Using TROPOMI (TROPOspheric Monitoring Instrument) satellite data, super-emitters

were detected for coal mining, oil and gas production regions, and along the major gas transmission pipelines (Schuit et al., 2023). The majority of detected super-emitters is related to urban areas (35% of detected super-emitters), with a possible large contribution from landfills (Schuit et al., 2023), where no direct samples of $\Delta^{13}CH_3D$ and $\Delta^{12}CH_2D_2$ have been taken so far.

Comparing locations of field samples and a map of anthropogenic CH$_4$ emissions, based on EDGAR v8.0

inventories, there is a considerable deficiency in measurements of doubly substituted isotope ratios from numerous locations with elevated CH$_4$ emissions (figure 8). No samples have been analysed from regions with significant CH$_4$ emissions, like Central Africa, southwestern South America, India, Pakistan, western China, New Zealand, and Indonesia. There is no data from the EDGAR database for certain areas, such as Siberia and Canada, where increased anthropogenic emissions can occur as well.

Furthermore, sampling should be conducted in regions with notable natural emissions, such as wetlands and internal freshwaters, including thawing permafrost.

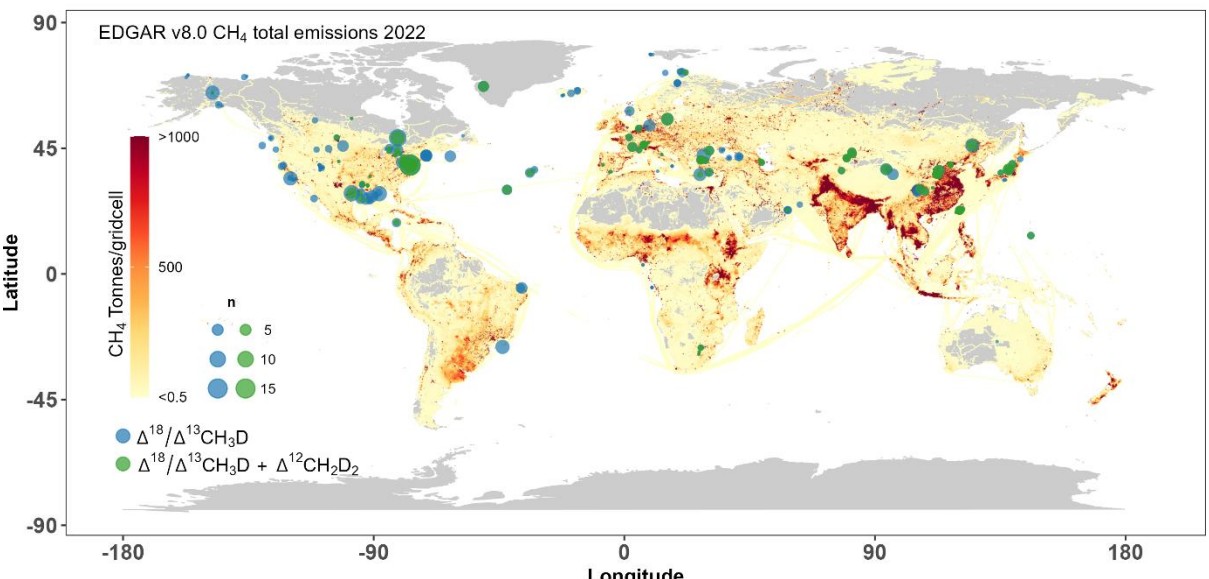

**Figure 8.** Global locations of collected field samples for doubly substituted isotope measurement (blue and green circles) overlaid on an estimate of the total CH$_4$ annual emission rates for year 2022 from the EDGAR v8.0 inventory.

## 5. Data availability

Data may be accessed from the following DOI:
Defratyka, S. M., Fernandez, J. M., and Arnold, T.: Methane doubly substituted (clumped) isotopologues database, CEDA Arch., https://dx.doi.org/10.5285/51ae627da5fb41b8a767ee6c653f83e6, 2025.

## 6. Conclusions

This study presents a compilation of $\Delta^{13}CH_3D$ and $\Delta^{12}CH_2D_2$ measurements from field samples and laboratory experiments, from results published between 2014 and 2025, by numerous laboratories. The database is designed for utilization by the geochemistry and atmospheric science communities. The database of doubly substituted isotope ratios comprises 1475 data records from 75 peer-reviewed articles (figure 2a and 4). Of this data, 53% of the database entries report only $\Delta_{18}$ or $\Delta^{13}CH_3D$, which can hinder data interpretation, especially for microbial, abiotic or mixed samples, when used without any additional tracer (Chowdhury et al., 2024; Douglas et al., 2017; Giunta et al., 2019; Gruen et al., 2018; Thiagarajan et al., 2020; Warr et al., 2021a; Young et al., 2016, 2017). For field samples, 40% of the data records come from natural gas, mostly from the basins in the US and China. Samples collected from lakes contribute 75% of microbial terrestrial samples. At the current state, there is a limited representation of samples coming from wetlands and agriculture sources and there is no representation of directly sampled waste sector (figure 2b).

As our ability to measure doubly substituted isotopologues of $CH_4$ in the atmosphere improves, a commensurate effort to improve our understanding of source signatures is needed in order to make the very most of these measurements in understanding the global atmospheric $CH_4$ budget. Studies should focus on the main emission sectors to the atmosphere, in particular on underrepresented sectors such as agriculture (e.g., ruminants, manure, rice cultivation), wetlands (including polar), waste and biomass burning. Also new field campaigns should focus on areas with increased $CH_4$ emissions, including super-emitters. An additional effort is also required to provide more ambient air background samples, ideally from remote, clean air sites. To better understand $CH_4$ sinks, more experiments focused on photochemical oxidation by OH and Cl must also be conducted.

**Supplement link** (link for excel spreadsheet with Tables S1-S6, given by editor of ESSD)

**Author contributions**

**Conceptualization:** SMD, JMF, TA; **Investigation and data curation:** GAA, GD, PMJD, DLE, GE, TG, MAH, ANH, NH, VI, JJ, JHK, JL, EL, WL, JL, LHL, JL, LO, SO, JR, TR, BSL, MS, JS, GTV, DTW, EDY, NZ; **Formal analysis:** SMD; **Visualization:** SMD; **Writing (original draft preparation):** SMD; **Writing (review and editing):** SMD, JMF, TA, GAA, GD, PMJD, DLE, GE, TG, MAH, ANH, NH, VI, JJ, JHK, JL, EL, WL, JL, LHL, JL, LO, SO, JR, TR, BSL, MS, JS, GTV, DTW, EDY, NZ.

**Competing interests**

The authors declare that they have no conflict of interest.

**Acknowledgements**

For the purpose of open access, the author has applied a Creative Commons Attribution (CC BY) licence to any Author Accepted Manuscript version arising from this submission. We gratefully acknowledge the authors and researchers whose previously published work was used for this data aggregation. The data compiled herein are derived from and built upon findings reported in peer-reviewed scientific literature. All original sources have been cited appropriately in the accompanying references.

Funding for this work came from the UKRI NERC POLYGRAM project NE/V007149/1 (www.polygram.ac.uk), the EURAMET 21GRD04 isoMET project and the NPL Director's Fund. The 21GRD04 isoMET project has received funding from the European Partnership on Metrology, co-financed from the European Union's Horizon Europe Research and Innovation Programme and by the Participating States. This article is not a product of the U.S. Department of Energy. Views and opinions expressed in this article are the authors' own and do not represent those of the United States Government.

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
