# Peer review of "Global inventory of doubly substituted isotopologues of methane ( $\Delta^{13}CH_3D$ and $\Delta^{12}CH_2D_2$ )"

_Earth System Science Data, 2025_

## Author Comment (AC1)

**Authors' response to Anonymous Referee #1**

**R:** This paper seeks to organize all measurements of methane clumped isotope measurements published to date. The introduction, explanations, and discussions are complete and it was a pleasure to read.

I have only two larger suggestions for the authors. I leave it to the authors to decide if they want to follow these suggestions.

**A:** We sincerely thank the reviewer for their positive and encouraging feedback. Our goal was to provide a comprehensive and accessible synthesis of methane clumped isotope measurements, and we are grateful that this effort was well received.

We also appreciate the reviewer's two larger suggestions and other minor comments. We have carefully considered them and provide detailed responses below.

**R:** First, I would suggest providing plots of dD and d13C vs the 13CD and D2 measurements as well as dD vs. d13C – I would provide in the background prior databases for bulk so the data here can be compared to that. This is a way of showing whether the data set assembled here compares well to prior data measured or if there are gaps. Second, I think it is important to consider the clumped data in general in the context of prior bulk isotope measurements such that interpretations of clumping are considered in conjunction with standard measurements.

**A:** We appreciate the reviewer's suggestion to put our study in context of previous bulk isotope measurements. To address this, we have compared  $\delta^{13}\text{C-CH}_4$  and  $\delta\text{D-CH}_4$  values from our database with major bulk isotope databases: Sherwood et al. (2021)and Menoud et al. (2022). The comparison is structured by methane source type (i.e. group type in the database): fossil fuels versus thermogenic; microbial; pyrogenic. We added an additional paragraph, plot and table for better data visualisation. For simplification, on the plot only our database and Menoud et al. (2022) are compared.

Changes in the manuscript: lines 234-253: "To verify if the compiled data compares well with previous studies, figure 1 and table 1 present bulk isotopes from this database in the reference to previously reported  $\delta^{13}$ C-CH4 and  $\delta$ D-CH4 (Menoud et al., 2022a; Sherwood et al., 2021). Across compared group types, our additional bulk isotope ratio data fall within the established ranges. Fossil fuel and thermogenic source signatures overlap, however, they are not strictly equivalent. Thermogenic CH4 in our dataset is slightly enriched ( $\delta^{13}$ C-CH4:  $-39.0 \pm 9.6\%$ ;  $\delta$ D-CH4:  $-169.2 \pm$ 41.9‰), compared to fossil fuel. For the comparison, only terrestrial microbial (e.g., agriculture, lakes, wetlands) from this database is compared with previously compiled data and shows strong agreement with the range of previous microbial samples, with depleted  $\delta^{13}$ C-CH4 and  $\delta$ D-CH4 values ( $\delta^{13}$ C-CH4: -62.9 ± 13.2%;  $\delta$ D-CH4: -298.1 ± 47.7%). Pyrogenic methane, though represented by only two samples in the new database, shows  $\delta^{13}$ C-CH4 and  $\delta$ D-CH4 values consistent with previous studies. This alignment supports the representativeness of our inferred doubly substituted CH₄ isotopologues ratio source signatures for use alongside the bulk isotope ratios in global modelling of the CH₄ budget. Our database also provides further additional measurements of the bulk isotopes to aid in further work to refine the source signatures  $\delta^{13}$ C-CH4 and  $\delta D$ -CH4.

**Figure 1.** Database entries plotted as  $\delta^{13}$ C-CH4 versus  $\delta$ D-CH4 alongside the Menoud et al., 2022a database. Error bars are taken from original studies. ff: fossil fuels, Tr: thermogenic, M: microbial, Pr: pyrogenic.

**Table 1.** Comparison across the three databases of  $\delta^{13}$ C-CH4 and  $\delta$ D-CH4 by group type. The mean value is reported with  $\pm$  1 standard deviation, and minimum and maximum values in brackets.

|                        | $\delta^{13}$ C-CH 4 |        |                 | δD-CH₄  |        |                  |
|------------------------|---------------------------------|--------|-----------------|---------|--------|------------------|
|                        |                                 | median |                 |         | median |                  |
| Group type             | samples                         | (‰)    | mean (‰)        | samples | (‰)    | mean (‰)         |
| fossil fuels           |                                 |        | -45.5 ± 9.1     |         |        | -185.5 ± 38.7    |
| Menoud et al. 2022     | 707                             | -44.2  | [-82.1; -18.3]  | 394     | -185.3 | [-355.0; -63.8]  |
| fossil fuels           |                                 |        | -44.9 ± 10.6    |         |        | -196.1 ± 48.6    |
| Sherwood et al. 2021   | 9477                            | -43.0  | [-87.0; -14.8]  | 3371    | -191.7 | [-415.0; -62.0]  |
| thermogenic            |                                 |        | $-39.0 \pm 9.6$ |         |        | -169.2 ± 41.9    |
| Defratyka. et al. 2025 | 309                             | -38.3  | [-73.0;-21.6]   | 309     | -159.7 | [-300.2; -100.8] |
| microbial              |                                 |        | -58.5 ± 8.5     |         |        | -309.7 ± 50.4    |
| Menoud et al. 2022     | 471                             | -58    | [-96.1; -36.5]  | 187     | -307.1 | [-472.0; -93.2]  |
| microbial              |                                 |        | $-61.6 \pm 6.9$ |         |        | -304.0 ± 36.6    |
| Sherwood et al. 2021   | 131                             | -62.4  | [-79.6; -45.5]  | 20      | -304.0 | [-358.0; -205.0] |
| microbial              |                                 |        | -62.9 ± 13.2    |         |        | -298.1 ± 47.7    |
| Defratyka. et al. 2025 | 120                             | -66.8  | [-77.7; 4.2]    | 120     | -294.7 | [-383.5; -90.5]  |
| pyrogenic              |                                 |        | -25.9 ± 7.7     |         |        | -176.7 ± 59.0    |
| Menoud et al. 2022     | 42                              | -27.2  | [-42.7;-9.0]    | 11      | -192.0 | [-285.0; -81.0]  |
| pyrogenic              |                                 |        | $-26.0 \pm 5.3$ |         |        | -21.8 ± 15.5     |
| Sherwood et al. 2021   | 29                              | -26.9  | [-33.4; -12.5]  | 4       | -208.0 | [-232.0; -195.0] |
| pyrogenic              |                                 |        | -27.7 ± 1.6     |         |        | -248.6 ±10.7     |
| Defratyka. et al. 2025 | 2                               | -27.7  | [-28.8;-26.5]   | 2       | -248.6 | [-256.1; -241.0] |

Once paragraph describing bulk isotopes comparison was added, lines 232-234 were slightly changed to improve the flow of reading. New lines 254-259: "The references included in the database of doubly substituted  $CH_4$  isotopologues comprise mostly peer-reviewed articles, with a smaller percentage from conference papers. The aggregated studies were carried out between 2014 and 2025 across 10 laboratories worldwide. As the aim of this study is to include all existing studies of doubly substituted isotopologue ratios, we also incorporated results from laboratory experiments, and of  $CH_4$  dissolved in water (i.e. in oceans, wetlands, and inland waters), which were not included in bulk isotopes databases."

R: Second, in the database, I strongly suggest providing the additional 'metadata' such as, for surficial samples, the dD of waters, d13C of CO2 etc. For experiments, such would also be

extremely helpful and, where known, the isotopic composition of the organic molecules provided (where relevant). For the thermogenic samples, I would suggest providing the gas compositions and isotopic composition of other molecules (where know). I know this will be annoying to do, but this is the kind of information that makes the database become extremely useful as, any study using this data, will likely need that as well. And so future authors will be stuck compiling this other information over and over again. I note, maybe this is provided, but I only saw a cell indicating what metadata exists.

**A:** We thank a lot to the reviewer for this valuable comment. We agree that including additional metadata could benefit the community and significantly facilitate future work. However, incorporating such diverse information would require substantial effort, particularly in designing an unified format suitable for integration, which is beyond the scope of the current study. For now, the "other tracers" column allows users to filter the dataset in a flexible way. Also, we aim to keep database "live" and updated it every few years. Potentially more additional effort can be done in the future to include remaining metadata in the database in the future. We added lines 270-271: "This parameter can be used to filter and group data for the further processing by database users".

\*\*\*\*

**A few minor comments:**

**R:** Line 101-102: clumping is only independent of bulk composition for an equilibrated system. It is a strong function of bulk composition for many non-equilibrium processes (mixing, chemical kinetics, distillation, etc.).

**A:** We thank the reviewer for this clarifying comment. We implemented necessary corrections in the text. In updated version we made small changes to improve the explanation: line 89-91 "This parameterization proves beneficial, as at thermodynamic isotopic equilibrium, the deviation in these isotopologue ratios from a purely random distribution is solely a function of temperature and it is independent from the bulk isotopic contents." And line 103: "Therefore, measurements of doubly substituted isotopologues can provide additional analytical..."

**R:** Line 113: In terms of history — there are older attempts to do methane clumping for 12CD4 and claims of exceptional values. Eiler 2007 summarizes this. Ma et al (2008 https://doi.org/10.1016/j.gca.2008.08.014) discussed the idea of the measurement. Tsuji et al. (2012 https://doi.org/10.1016/j.saa.2012.08.028) also attempted this and developed a method, but I wasn't applied to measurements of environmental samples as far as I recall.

**A:** We thank for bringing this historical context. We included additional sentence in the manuscript, to better present historical path toward measurements of  $\Delta^{12}CH_2D_2$  and  $\Delta^{13}CH_3D$ . In, lines114-116: "The first attempt to measure the rare  $CH_4$  isotopologues from the ambient air was presented by Mroz et al. (1989), with further methods development refined by Ma et al. (2008) and Tsuji et al. (2012)."

**R:** Line 379-383. I understood that there is an emerging understanding that for thermogenic methane, it likely forms out of clumped equilibrium but, at high enough temperatures, rapidly reequilibrates so it reflects peak formation temperature prior to expulsion. This is discussed as far as I remember it the cited papers from Dong, Eldridge, and Xie et al. This is a nuance, but is different from methane representing formation temperatures and formation in equilibrium but rather represents rapid kinetics of H exchange post methane formation and then quenching of

the reaction. I recommend checking those papers to verify what they said (or asking them as two are on this paper).

**A:** We appreciate a lot this valuable suggestion which allows to make the manuscript more accurate. In the new version in line 420: clarification: "implying at least partly kinetically-driven signatures" is added.

**R:** For figure 6, what are the catalytic equilibration samples that are +30 to+40%? Are those labeled experiments where 13CD was added to a sample then removed during equilibration to verify the catalyst was working? If so, I might not include as they are spiked experiments. I would in general avoid including anything in which labels were added.

A: We thank the reviewer for highlighting this point. For figure 6 (in updated manuscript it is figure 7), those high values of  $\Delta^{13}$ CH3D come from initial isotopic signature of the methane (time of experiment =0 or close to 0), thus this value shows isotopic signature before the start of equilibration We decided to leave those values in the manuscript, as they are starting points of equilibration experiments. For clarification, we added an explanation in the manuscript: line 569-571: "The outliers for catalytic equilibration come from the sample measured at the beginning of the experiment, when equilibration on the catalyst did not start yet." Also, based on the reviewer comment, we decided to remove from figure 3 and 5 datapoints from deuterium-enriched substrate, as obtained results do not appear in the nature. We added explanation in caption of figure 3 (line 362-364) and 5 (line 390-393): "Laboratory experiments with deuterium-enriched water substrate (Gruen et al., 2018; Li et al., 2024, 2025a; Taenzer et al., 2020) are not included as they do not appear under normal incubation or environmental conditions." We also added an additional explanation to highlight the need of careful reinterpretation of experiments with deuterium-enriched substrate, line 369-373: "Notably, Gruen et al., (2018), Li et al., (2024, 2025a), and Taenzer et al., (2020), carried out incubations with deuterium-enriched substrate to explore mechanisms behind combinatorial effects. Thus, observed clumped isotopologues do not represent the isotopic values of natural-occurring microbial CH4 and should be carefully reinterpreted."

---

## Author Comment (AC2)

**Authors' response to Anonymous Referee #2**

**R:** The Defratyka et al. manuscript describes the distribution of doubly substituted isotopologues in exospheric reservoirs and their relevance in constraining the sources of atmospheric methane. The paper is important in that describes how methane clumped isotopes can potentially help in the quantification of atmospheric sources which are of key importance to atmospheric scientists working to model the source and sink of key greenhouse gases, including methane.

Two challenges to this approach (as described in the introduction) have been the incorrect assignment of kinetic isotope effects in sink reactions, and also assumptions regarding the source signatures (Line 122). In order to correct this, the authors have compiled a database of 1500 previously published clumped isotope results. Because the equilibrium distribution of isotopologues is temperature dependent, the sources can be inferred based upon expected source temperatures, although this also has some caveats (Line 97).

Of the field-sampled sources, thermogenically derived methane is the most common, as has been verified through traditional gas measurements of cold seeps in recent decades (e.g. stable carbon isotopes of methane, methane/ethane ratios, etc.). Similar approaches have been taken to determine biogenic end-member distribution in cold seeps. As Defratyka et al. concede in this manuscript (line 331), abiotic methane is quite poorly constrained. The sources and distribution of abiotic methane are adequately described in this paper (line 405). Nonetheless, the difficulties in ascertaining the abiotic component are not that straightforward, given that the methane concentrations are quite low in high-temperature gases such as those in submarine hydrothermal vents, geothermal wells, and fumoroles. In these cases, CO2 gas and, in some cases nitrogen gas, are major constituents while methane may only be a couple percent of the total gas content (as opposed to cold seeps where methane often comprises over 90% of the total gas content). As such, abiotic methane derived at high temperatures can easily be contaminated by other sources, such as thermogenic methane gas released during pyrolysis of organic matter in surrounding sediment and, as reviewer, it is of my personal opinion that methane ascribed as being of purely abiotic origin be viewed with some caution.

As described by the authors, the net sink for methane in wetlands and marine systems is either AOM or AeOM. The kinetic isotope effects associated with each process are unique and help distinguish between the two. On the other hand, tropospheric methane removal through OH reduction seems to have only a minimal effect on the remaining CH4.

This is the most complete compilation of doubly substituted methane isotopologues and the global map of sample distribution covers an impressive global range. Nonetheless, the sample density is patchy (fig. 7) and there are vast areas around the globe that remain unsampled, including some areas of very high methane emission. Given that data is lacking, and sampling is skewed towards some areas more than others, it is possible that further refinements in global sample distribution will change the distribution of clumped isotope results (fig. 5).

All in all, this data description paper is well written and describes the potential for clumped isotope studies to be applied to methane source determination in future studies. As such, I recommend this paper for publication, and also look forward to ongoing work which provides a more complete data set in the future.

**A:** We thank the reviewer for their review and positive comment. We also value that the difficulties of source attribution and kinetic isotope effects have been highlighted by the reviewer. We are also grateful for mentioning difficulties in detecting abiotic CH4, particularly in high-temperature

environments, where low methane concentrations and co-emitted gases provide analytical difficulties. It shows potential direction for more detailed field studies of abiotic  $CH_4$ . As we discussed in the manuscript, we also see the need of that future dataset extensions to enhance spatial and source coverage. Overall, we are grateful that the reviewer appreciates the current data compilation and they are enthusiastic to see further research of clumped isotopologues of  $CH_4$ .